# Effect of advanced periodontal self-care in patients with early-stage periodontal diseases on endothelial function: An open-label, randomized controlled trial

Ayako Okada[1,2], Takatoshi Murata[1]*, Khairul Matin[3,4], Meu Ariyoshi[1,4], Ryoko Otsuka[1], Mamiko Yamashita[1], Masayuki Suzuki[5], Rumi Wakiyama[5], Ken Tateno[5,6], Megumi Suzuki[7], Hitomi Aoyagi[8], Hiromi Uematsu[8], Akiko Imamura[8], Miki Kosaka[1,9], Tomoko Mizukaki[10], Tsutomu Sato[11,12], Hiroshi Kawahara[5], Nobuhiro Hanada[1]

1 Department of Translational Research, Tsurumi University School of Dental Medicine, Yokohama, Japan, 2 Department of Operative Dentistry, Tsurumi University School of Dental Medicine, Yokohama, Japan, 3 Endowed Department of International Oral Health Science, Tsurumi University School of Dental Medicine, Yokohama, Japan, 4 Division of Oral Health Sciences, Graduate School of Medical and Dental Sciences, Department of Cariology and Operative Dentistry, Tokyo Medical and Dental University, Tokyo, Japan, 5 Department of Dental Anesthesiology, Tsurumi University School of Dental Medicine, Yokohama, Japan, 6 Department of Anesthesiology, Saitama Medical University Hospital, Iruma-gun, Japan, 7 Department of Dental Hygiene, The Nippon Dental University College at Tokyo, Tokyo, Japan, 8 The Nippon Dental University Hospital, Tokyo, Japan, 9 Department of Dentistry, Tokyo Children Rehabilitation Hospital, Tokyo, Japan, 10 Department of Oral and Maxillofacial Surgery, St. Marianna University School of Medicine Kawasaki Municipal Tama Hospital, Kawasaki, Japan, 11 Division of Basic Medical Science, Tokai University School of Medicine, Isehara, Japan, 12 Louis Pasteur Center for Medical Research, Kyoto, Japan

* murata-ta@tsurumi-u.ac.jp

**Data Availability Statement:** The data contain potentially sensitive patient information. The Ethics Committee of the Tsurumi University School of

## Abstract

Although a significant association between periodontal disease and atherosclerotic cardiovascular disease has been reported, their cause-to-effect relationship remains controversial. This randomized controlled clinical trial aimed to investigate the effect of advanced self-care on atherosclerotic cardiovascular disease-related vascular function markers flow-mediated brachial artery dilatation (FMD) and serum asymmetric dimethylarginine (ADMA) level in patients with early-stage periodontal disease. The study was designed as a parallel group, 3-month follow-up, open-label, randomized controlled trial. The control group received standard care for periodontal diseases, whereas the test group additionally applied disinfectant using a custom-fabricated prescription tray for advanced self-care twice a day. Overall, 110 patients provided data for FMD and serum ADMA level. No significant improvements in FMD were observed in the control (mean increase, −0.1%; 95% confidence interval [CI], −1.0–0.8; P = 0.805) or test (mean increase, −0.3%; 95% CI, −1.1–0.4; P = 0.398) group. No significant changes in serum ADMA levels were observed (mean reduction, 0.01 µmol/L; 95% CI, −0.00–0.02; P = 0.366 and mean reduction, 0.00 µmol/L; 95% CI, −0.01–0.01; P = 0.349, respectively). No significant between-group differences were found in FMD (mean difference, −0.2%; 95% CI, −1.4–0.9; p = 0.708) or serum ADMA levels (mean difference, 0.01 nmol/L; 95% CI, −0.00–0.03; p = 0.122). Significant improvements in the average probing pocket depth were observed in the control and test groups. The

Dental Medicine prohibits the preservation of any data set in a public repository (even if the data are de-identified), the preservation of hard copy data in an unlocked cabinet, the preservation of electronic data in a computer connected to the internet. All the subjects provided informed consent on the condition of our adherence to IRB guidelines. The datasets generated and/or analyzed during the current study are available from Dr. Hidenori Yamada (yamada-h@tsurumi-u.ac.jp) or Dr. Ayako Okada (okada-a@tsurumi-u.ac.jp) on reasonable request. The Ethics Committee of the Tsurumi University School of Dental Medicine 2-1-3, Tsurumi, Yokohama, 230-8501 Japan Email Address: kyoken@tsurumi-u.ac.jp.

**Funding:** This work was supported by the Japan Society for the Promotion of Science (JSPS) [Grant numbers 16K20704 (MA), 17K12031 (AO), 17K11688 (KM), 18K09926 (NH), 19K10471 (TM), 19K19339 (RO)] and the SECOM Science and Technology Foundation [Grant number 2018.09.10 No. 1 (NH)]. Japan Society for the Promotion of Science: https://www.jsps.go.jp/ SECOM Science and Technology Foundation: https://www.secomzaidan.jp/ The funders had no role in study design, data collection and analysis, decision to publish, or preparation of the manuscript.

**Competing interests:** Dr. Khairul Matin and Dr. Nobuhiro Hanada received a research grant from Medoc International Co. Ltd. Dr. Nobuhiro Hanada received a research grant from Shiken Corp. This does not alter our adherence to PLOS ONE policies on sharing data and materials. There are no other competing interests to declare. Medoc International Co. Ltd.: http://www.medoc.co.jp/company Shiken Corp.: https://www.shiken-jp.com.

bleeding on probing score in the test group was significantly reduced, while that in the control group was reduced, although not significantly. Periodontal care for a 3-month duration did not provide better endothelial function although improvements of periodontal status in patients with early-stage periodontal diseases. This trial is registered in UMIN Clinical Trials Registry (www.umin.ac.jp/ctr/; ID: UMIN000023395).

## Introduction

Although a significant association between periodontal disease and atherosclerotic cardiovascular disease (ACVD) has been reported, the cause-to-effect relationship between them remains controversial [1]. The entry of oral bacteria, including periodontal pathogens and/or their products into the bloodstream, is common regardless of the periodontal status [2]. The immune response following persistent bacteremia from periodontal lesions may lead to ACVD [3]. Furthermore, it has also been suggested that inflammatory mediators attributable to periodontal diseases are associated with ACVD [4–6]. However, the verification of the hypothesis and identification of periodontal care effectiveness against ACVD require well-designed interventional studies.

Endothelial dysfunction occurs in the early stages of atherosclerosis and can be assessed by measuring the flow-mediated dilatation (FMD) of the brachial artery [7, 8]. Nitric oxide (NO), which is synthesized by vascular endothelial cells, is a primary contributor to FMD [7]. Noninvasively-measured FMD is considered a marker of vascular damage and predictor of future cardiovascular events [7–9]. Therefore, FMD appears to be an appropriate surrogate marker of periodontal care effectiveness against ACVD.

Observational studies have shown that periodontal diseases are associated with FMD [10]. Furthermore, some interventional studies without controls reported that the treatment of severe periodontitis improved FMD [11–14]. Intensive periodontal treatment significantly improved FMD in a subject without systemic diseases in one randomized controlled clinical trial (RCT) [15]; whereas, no significant effects were observed in a patient with stable coronary artery disease in another RCT [16]. It was reported that intensive nonsurgical periodontal treatment significantly improved FMD in patients with hypertension in a recent RCT [17]. The relationship between the periodontal condition and FMD appears to be affected by complex factors, including severity of the periodontal condition, systemic condition, and contents of the treatment. Further RCTs are required to indicate the causative association between periodontal diseases and specific ACVD events and the usefulness of periodontal care to improve endothelial function.

Periodontal diseases are prevalent worldwide, with an estimated large majority of cases ranging in severity from the initial stages to more advanced conditions [18]. Bacteremia, which can induce ACVD, routinely occurs even in the early stage of periodontal disease [2, 19]. Research findings covering the severity may make a major impact on patient behavior for ACVD prevention. Daily self-care and professional care are important for periodontal treatment and prevention. Previous studies have shown that advanced periodontal self-care using a custom-made tray application of a disinfectant effectively decreases periodontal tissue inflammation [20–22]. Therefore, we hypothesized that advanced self-care improves endothelial function. This randomized clinical trial aimed to investigate the effect of advanced self-care on ACVD-related vascular function markers. Our null hypothesis was that there was no difference in FMD between advanced self-care and standard care in patients with early-stage periodontal disease.

## Materials and methods

### Trial design

This parallel group (1:1), 3-month follow-up, open-label, RCT was conducted at the Tsurumi University Dental Hospital in Yokohama, Japan, and the Ariyoshi Dental Clinic in Tokyo, Japan, between August 2016 and April 2018 by a single team composed of dentists, dental hygienists, and nurses. The dental hospital and clinic were well equipped to provide standard care for the initial periodontal preparations. All participants chose the trial site at their convenience.

This study was conducted in accordance with the ethical principles of the Declaration of Helsinki. The study protocol was approved by the Ethics Committee of the Tsurumi University School of Dental Medicine (No. 1413) and registered in the University Hospital Medical Information Network Clinical Trials Registry (UMIN-CTR: UMIN000023395). All participants provided written consent before the study began.

### Participants

Participants were recruited between August 2016 and November 2016 from various sources, including advertisements in social networking services, the Tsurumi University Dental Hospital, and two private dental clinics in Tokyo, Japan. Between November 2016 and February 2017, 150 volunteers were assessed for eligibility in accordance with the inclusion and exclusion criteria. The inclusion criteria were an age range of 20–70 years, having $\geq$20 functioning teeth, and diagnosis of chronic periodontitis with $\geq$2 sites with bleeding on probing (BOP) or a probing pocket depth (PPD) of $\geq$4 mm at $\geq$1 site. The exclusion criteria were difficulties traveling alone to the institution, the need for periodontal surgery and/or prosthodontic treatment, the presence of an untreated carious cavity, the presence of a partially impacted tooth, having used systemic antibiotics within the previous 3 months, the regular use of medications for any other chronic diseases, the presence of diseases requiring higher-priority treatment than periodontitis, and participation in another clinical trial during the study period.

### Interventions

All periodontal patients received standard care for periodontal diseases under local anesthesia if required, including same-day full-mouth scaling between April 2017 and January 2018. Patients who underwent standard care alone were assigned to the control group. Eleven care providers (Authors: R.O., M.S., H.A., H.U., A.I., Non-authors: refer to Acknowledgments; T.K., E.S., N.A., Y.U., M.E., J.T.) with >5 years of experience in dental practice discussed and standardized the procedures of the care before the trial began. Patients were randomly assigned to each care provider who conducted their care in a similar fashion using an ultrasonic scaler (Varios970; Nakanishi, Tochigi, Japan), hand instruments (FP scaler; Feed, Yokohama, Japan), and dental floss (Reach No-waxed; Johnson & Johnson, Tokyo, Japan) with a dental plaque-disclosing agent (Merssage PC Pellet Blue; Shofu, Kyoto, Japan) to visualize and remove dental plaque biofilms. After the complete removal of the plaque, all teeth surfaces were polished with a polishing paste (PTC Paste Fine/PTC Paste Regular; GC, Tokyo, Japan) using a rotating rubber cup (FP rubber; Feed, Yokohama, Japan) and/or rotating brush (FP profy brush; Feed, Yokohama, Japan). Complete plaque removal was confirmed with the repetitive application of the disclosing agent. No time restriction was imposed on the procedure. One dentist (T.M.) who was not a part of the care-providing team confirmed the fulfillment of the standardized care program and subsequently provided nutritional and exercise guidance in addition to basic oral hygiene instructions. Each patient was given the same toothpaste

(Check-up standard; Lion, Tokyo, Japan) and toothbrush (Ci 202 premium; Ci Medical, Hakusan, Japan) every month during the trial and was instructed to perform standard self-care at least twice (in the morning and at bedtime) a day.

For patients assigned to the test group, the use of hypochlorous acid water, an electrolyzed disinfectant for dental use (MeDeSPro; Medoc International, Tokyo, Japan), with a custom-made prescription tray was introduced as an extra intervention in addition to standard self-care. The patients were instructed on the use of these materials after standard care for periodontal diseases under the guidance of one dentist (T.M.). The process of manufacture of the tray have been described previously [23] and is described below.

The inside of the tray was covered with gauze. Each patient was instructed to fill the tray with approximately 3 mL of disinfectant and then apply it to the tooth and gingival surfaces for 3 min after each standard self-care (twice a day). They were instructed not to eat or drink within 30 min after using disinfectants. We considered a 70% implementation rate as successful advanced periodontal self-care.

We asked all patients in both groups to keep a record of the frequency of oral self-care, general health condition, and drug being taken in a diary throughout the trial.

## Custom-made tray

Maxillary and mandibular dentition impressions were taken from each patient, and a dental working model using dental plaster (New Plastone II; GC, Tokyo, Japan) was prepared. The custom-made trays were manufactured at a contract dental laboratory (Shiken, Tokushima, Japan or Oral Bio Design, Shiga, Japan). Gauze and polypropylene sheets stacked in pairs were vacuum-adapted to each model using a vacuum-forming machine (Biostar; Scheu Dental, Iserlohn, Germany). Each tray was trimmed to an extension of approximately 5.0 mm above the maximum convexity of the jawbone, and inner gauze was trimmed to an extension of approximately 2.0 mm above the gingival margin (S1 Fig).

## Outcome

The primary outcome was the occurrence of changes in FMD between baseline and 3 months after the start of the intervention. The change in serum asymmetric dimethylarginine (ADMA; an endogenous NO synthase inhibitor) levels was considered a secondary outcome. Elevated serum ADMA levels are thought to impair endothelial function and, promoting atherosclerosis as a result [24].

## Sample size

We employed a priori power analysis for sample size calculation. Based on a previous study [15], we calculated that a minimum sample size of 90 was needed to detect a 1% difference in FMD between the two groups, with a standard deviation of the mean difference of 1.67% at a 2-sided α error of 0.05 and 80% power. Assuming a 20% dropout rate, 110 subjects were enrolled.

## Randomization

The eligible patients were allocated randomly to the test group or control group via the envelope method stratified by the smoking habit, which is a common risk factor for ACVD and periodontal disease. Almost all participants wished to be allocated to the test group. Therefore, we used the following procedures for the allocation. We prepared 110 brown, standard, and non-labeled envelopes (29 for smokers and 81 for nonsmokers) containing a label with an

Arabic number (1 to 29 for smokers and 1 to 81 for nonsmokers). All the researchers confirmed that no one could identify the number from the outside of each sealed envelope. We explained to all patients during eligibility assessment that the odd- and even-numbered labels were allocated to the control and test groups, respectively, and the custom-made tray would be manufactured for all control patients who desired one after their own trial. After eligibility assessment, each patient selected as a study subject made an appointment for random allocation at their convenience. Patients drew the envelope by themselves from the "smoker" or "nonsmoker" labeled box as required, and opened the envelope in front of us. Balloting was done on a first-come, first-served basis. The allocated Arabic number was particular for each subject, and the number was never used again. No one declined trial participation because of the allocation.

## Periodontal examination

A single examiner (A.O.) collected the following clinical data from six sites (mesio-buccal, mid-buccal, disto-buccal, mesio-lingual, mid- lingual, and disto-lingual) around each tooth using a color-coded periodontal probe (PO-9; Nippon Shiken, Tokyo, Japan). PPD was measured from the free gingival margin to the base of the pocket. The BOP was considered positive if a site bled within 20 s after gentle probing (25 g probing force). The average PPD and percentage of BOP sites (BOP score = number of sites with BOP/total number of sites × 100) were calculated for each patient.

## FMD of the brachial artery

FMD was assessed using a high-resolution ultrasonography system (Unexef18G; UNEX, Nagoya, Japan) in an air-conditioned room. Each patient was instructed to refrain from physical activity, caffeine-containing foods, and smoking for at least 2 h before the examination. Blood pressure and heart rate were measured before measuring FMD. The examination began after a 10-min rest with the patient in the supine position and the arm placed comfortably. The left brachial artery was scanned with an ultrasound probe at 10 MHz for longitudinal and transverse windows. After recording the baseline image, the cuff that was placed distal to the ultrasound probe was inflated to 50 mmHg above systolic blood pressure for 5 min and then deflated. The post-deflation arterial image was recorded for 2 min. FMD was defined as the maximal percentage change in vessel diameter from the baseline value. All measurements were performed by a single operator (A.O.).

## ADMA measurement

Right antecubital vein blood samples were placed into collection tubes containing serum separator gel and immediately centrifuged. The supernatants were frozen rapidly at -30˚C for later analysis. Serum ADMA levels were measured at a contract laboratory (SRL, Tokyo, Japan) by high-performance liquid chromatography equipped with a fluorescence detector for excitation at 348 nm and emission at 450 nm with an octadecyl-silica column using derivatization with *o*-phthalaldehyde for fluorescent determination.

## Statistical analysis

All analyses were performed based on the intention-to-treat principle. Missing data were managed using the baseline-observation-carried-forward approach in accordance with the existing guidelines [25]. A per-protocol analysis was also performed without imputation. The patients who took antibiotics during the trial were excluded from per-protocol analysis. Data are

expressed as means and 95% confidence intervals (CIs) unless otherwise specified. The distribution of each continuous variable was checked for normality using the Shapiro–Wilk test. Levene's test was performed for homogeneity of the variances when normality was confirmed. Paired statistical tests (paired Student's *t*-test or Wilcoxon signed-rank test) were used to compare the difference in parameters obtained at baseline and endpoint in the same group, whereas unpaired tests (Student's *t*-test, Welch's *t*-test or Mann–Whitney U test) were used to compare differences between the control and test groups. The correlation between two parameters was assessed using Spearman's rank correlation coefficient. All tests were 2-tailed, and values of $p < 0.05$ were considered statistically significant. All analyses were performed using JMP Ver. 12 (SAS Institute, Cary, NC, USA).

## Results

### Participant flow

The flow diagram of the trial is presented in Fig 1. Forty volunteers were excluded from the trial. The remaining 110 patients were allocated randomly to the control (n = 56) and test (n = 54) groups. The baseline characteristics of the patients were similar between the groups (Table 1). Forty-three patients in the test group achieved ≥70% advanced self-care implementation (mean, 85.1%; minimum, 34.7%; maximum, 100%). Control patients received the intervention immediately after allocation, whereas the test group began the intervention 2 weeks after allocation because of the need for custom-made tray manufacture. No severe adverse events were observed during the trial.

### Vascular function

An improvement in FMD of at least 1% difference was not achieved between the groups. Table 2 presents the results of FMD and serum ADMA levels obtained at the endpoint and improvement by the intervention in the intention-to-treat analysis. The change in FMD at endpoint did not differ significantly between the control and test groups. No significant improvements in FMD were observed in the control or test groups. No significant between-group difference was detected (mean difference, −0.2%; 95% CI, −1.4–0.9; p = 0.708, unpaired *t*-test) (S2 Fig). The change in serum ADMA levels at endpoint differed significantly between the control and test groups. No significant improvements in serum ADMA levels were observed in either control group or test group, with no significant between-group difference observed (mean difference, 0.01 nmol/L; 95% CI, −0.00–0.03; p = 0.122, Mann–Whitney U test) (S2 Fig). The result in the per-protocol analysis was also similar (S1 Table and S3 Fig). FMD was not correlated with serum ADMA levels (r = 0.095, p = 0.172).

### Periodontal condition

Table 3 presents the results of periodontal status obtained at endpoint and improvement by the intervention in the intention-to-treat analysis. The change in Mean PPD or BOP at endpoint did not differ significantly between the control and test groups. Significant improvements in the average PPD were observed in the control and test groups. The BOP score in the test group was significantly reduced, while that in the control group was reduced, although not significantly. No significant between-group differences were found in the improvement of the mean PPD (mean difference, −0.0 mm; 95% CI, −1.1–0.1; p = 0.820, Mann–Whitney U test) or BOP score (mean difference, −0.1%; 95% CI, −3.1–2.9; p = 0.955, Mann–Whitney U test). The result in the per-protocol analysis was also similar (S2 Table).

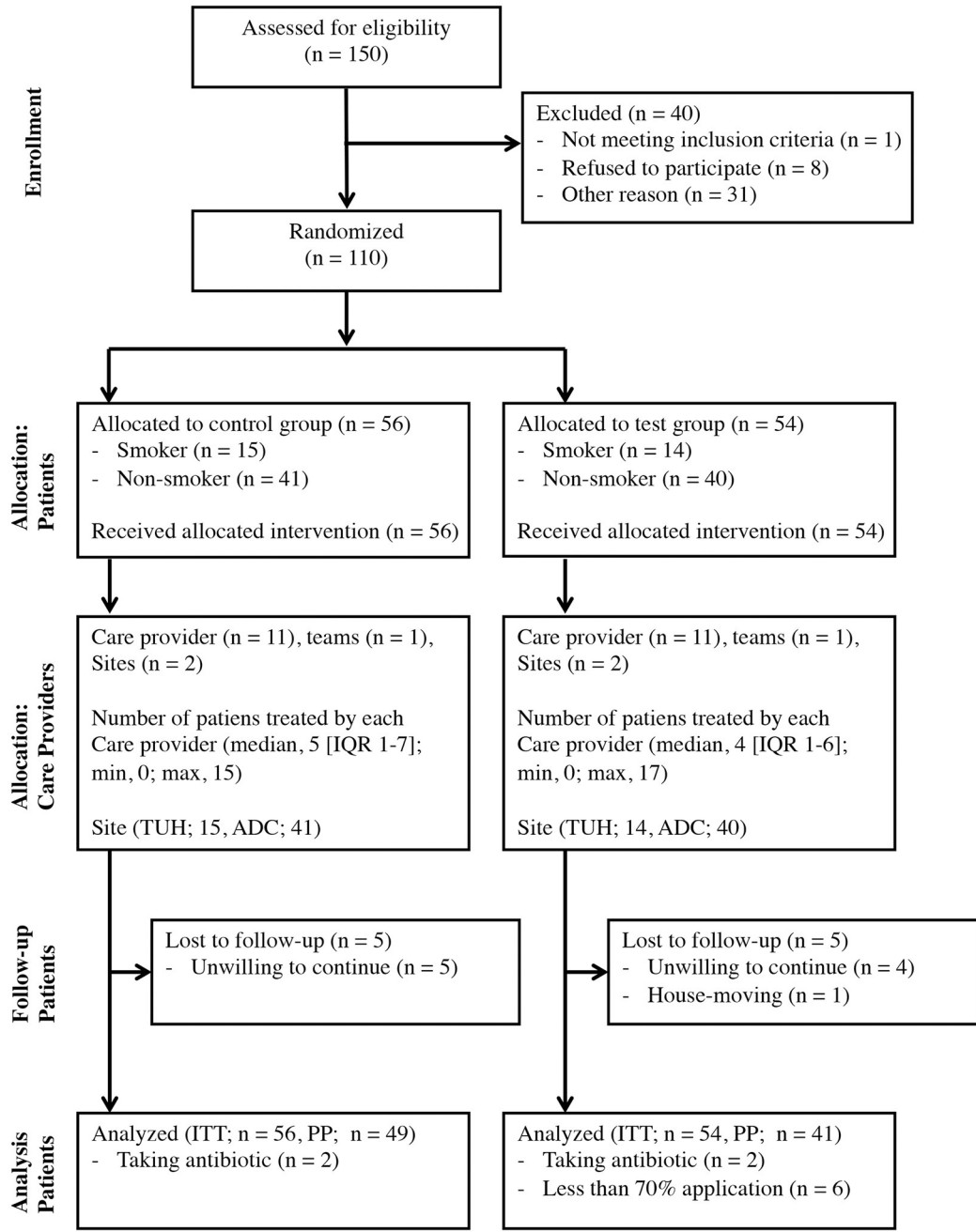

**Fig 1. Flow diagram.** IQR, interquartile range; TUH, Tsurumi University Dental Hospital; ADC, Ariyoshi Dental Clinic; ITT, intention-to-treat; PP, per-protocol.

## Harms

No study-related serious adverse events occurred in any of the study participants. Nonsurgical periodontal treatment is basically low-risk. None of the participants required any dental therapy during the study.

**Table 1. Baseline characteristics of the patients.**

| Characteristic | Control (n = 56) | Test (n = 54) |
|---|---|---|
| Age* (years), median (IQR) | 37 (29–43) | 38 (32–45) |
| Male sex, No. (%) | 30 (54) | 36 (67) |
| Smoker, No. (%) | 14 (25) | 13 (24) |
| Existing teeth (No.), median (IQR) | 28 (27–30) | 28 (28–29) |
| Average PPD (mm), median (IQR) | 2.1 (1.9–2.4) | 2.0 (1.8–2.1) |
| BOP score (%), median (IQR) | 10.8 (5.9–20.0) | 9.5 (4.6–15.9) |
| Systolic BP (mmHg), mean ± SD | 119.6 ± 10.9 | 120.4 ± 14.7 |
| Diastolic BP (mmHg), mean ± SD | 70.4 ± 9.3 | 73.4 ± 11.6 |
| Brachial artery diameter (mm), mean ± SD | 3.4 ± 0.8 | 3.6 ± 0.7 |
| FMD (%), mean ± SD | 5.9 ± 3.0 | 5.8 ± 2.9 |
| ADMA (nmol/L), median (IQR) | 0.36 (0.32–0.39) | 0.34 (0.30–0.37) |

IQR, interquartile range; No., number; PPD, periodontal pocket depth; BOP, bleeding on probing; SD, standard deviation; FMD, flow-mediated dilatation of the brachial artery; ADMA, serum asymmetric dimethylarginine level.

## Discussion

Advanced periodontal self-care for three months did not significantly improve FMD as compared to standard care. The findings were similar in the intention-to-treat and the per-protocol analysis. Therefore, our null hypothesis 'there was no difference in FMD between advanced self-care and standard care in patients with early-stage periodontal disease' was not rejected.

The immune response following persisting bacteremia from periodontal lesions reportedly leads to ACVD regardless of the severity of the periodontal condition [2, 3, 19]. In addition, the production of inflammatory mediators attributable to periodontal diseases might be associated with ACVD [4–6]. Reduced FMD is associated with ACVD risk and improves with risk-reduction therapy [26]. Consequently, the endothelial function has been defined as an "excellent barometer" of vascular health [27]. FMD assessing vascular endothelial function might be ideal for exploring factors associated with the improvement of vascular health. Based on these suggestions, several studies have described the efficacy of periodontal care for FMD [11–14].

**Table 2. Assessment of vascular function at 3 months after the start of the intervention.**

| | Endpoint | | | |
|---|---|---|---|---|
| | Control (n = 56) | Test (n = 54) | Mean difference (95% CI) | p-value |
| FMD (%), mean ± SD | 5.8 ± 2.4 | 5.5 ± 2.3 | −0.3 (−1.2–0.5) | 0.436 |
| ADMA (nmol/L), median (IQR) | 0.36 (0.33–0.38) | 0.33 (0.31–0.38) | −0.03 (−0.05–−0.01) | 0.008 |
| | Improvement | | | |
| | Control (n = 56) | | Test (n = 54) | |
| | Mean difference (95% CI) | p-value | Mean difference (95% CI) | p-value |
| FMD (%) | −0.1 (−1.0–0.8) | 0.805 | −0.3 (−1.1–0.4) | 0.398 |
| ADMA (nmol/L) | 0.01 (−0.00–0.02) | 0.366 | −0.00 (−0.01–0.01) | 0.349 |

P-values were calculated using the unpaired t-test (FMD) or the Mann–Whitney U test (serum ADMA level) for group differences at endpoint and the paired Student's t-test (FMD) or Wilcoxon's signed-rank test (serum ADMA level) for changes from baseline. FMD, flow-mediated dilatation of the brachial artery; ADMA, serum asymmetric dimethylarginine level; SD, standard deviation; IQR, interquartile range; CI, Confidence interval.

**Table 3. Periodontal status at 3 months after the start of the intervention.**

| | Endpoint | | | |
| --- | --- | --- | --- | --- |
| | Control (n = 56) | Test (n = 54) | Mean difference (95% CI) | p-value |
| Mean PPD (mm), median (IQR) | 1.9 (1.8–2.1) | 1.9 (1.8–2.0) | −0.1 (−0.2−−0.0) | 0.072 |
| BOP (%), median (IQR) | 9.7 (3.7–17.2) | 8.2 (3.5–15.4) | −2.2 (−6.1–1.7) | 0.686 |
| | Improvement | | | |
| | Control (n = 56) | | Test (n = 54) | |
| | Mean difference (95% CI) | p-value | Mean difference (95% CI) | p-value |
| Mean PPD (mm) | 0.2 (0.1–0.2) | <0.001 | 0.2 (0.1–0.2) | <0.001 |
| BOP (%) | 1.8 (−0.8–4.4) | 0.052 | 1.7 (0.1–3.2) | 0.041 |

The Mann–Whitney U test was used for comparisons between groups at endpoint. The Wilcoxon signed-rank test† was used for comparison with baseline. PPD, periodontal pocket depth; BOP, bleeding on probing; IQR, interquartile range; CI, Confidence interval.

However, a limited number of RCTs have been conducted. A few RCTs reported the effectiveness of intensive periodontal treatment [15, 17], whereas others did not identify significant changes [16]. The discrepancy among RCTs might be attributable to differences in systemic characteristics between patients. The characteristics were population-specific, such as the presence of coronary artery disease or hypertension or the absence of systemic disease. In addition, the severity of the periodontal disease and the contents of the treatment also appeared to differ. We set the inclusion and exclusion criteria such that they emphasized the systemic health and periodontal status of the subjects. Therefore, we considered the study population to have good systemic health and early-stage periodontal disease requiring no surgical periodontal treatment. Despite the improved periodontal conditions observed in both groups, no significant increase in FMD was demonstrated in this trial. Periodontal treatment might be effective for improving FMD in patients with more severe periodontal disease than in patients with early-stage periodontal diseases. The main contents of standard care at the clinic are biofilm removal. Employment of more effective methods for the biofilm removal may affect the improving FMD through delayed biofilm regrowth rates [28].

We also selected serum ADMA levels as a secondary outcome. ADMA is a naturally occurring endogenous inhibitor of NO synthase [29–31]. It reduces NO production; consequently, it can lead to endothelial dysfunction and cardiovascular events [24, 32–34]. We hypothesized that the inhibition of the entry of oral bacteria into the bloodstream through periodontal care would lead to FMD improvement through the inhibition of ADMA synthesis. However, no significant decrease in serum ADMA levels from baseline was observed in spite of significant difference between the groups at endpoint. No significant correlation between FMD and serum ADMA levels was demonstrated in this trial; however, an inverse association between FMD and plasma ADMA levels has been reported in young Finns [35] and in subjects at low cardiovascular risk [36]. The reason for the discrepancy between the trials may be the underlying factors characterizing the study population which may be affecting FMD and/or plasma ADMA levels. Further investigations may be required to identify the factors in healthy subjects.

When mentioning the association between the periodontal condition and ACVD, appropriate periodontal parameters should be selected carefully to ensure a precise evaluation of the periodontal status. PPD and BOP represent the degree of swelling in the gingiva and the existence of a lesion at the periodontal pocket base, respectively. Changes in PPD and the number

of BOP sites directly reflect changes in the periodontal status. Therefore, the mean PPD and number of BOP sites appeared to be appropriate indices for similar studies to portray an improvement and/or a deterioration of the periodontal status. The mean PPD and number of BOP sites improved in both study groups in this trial. On statistical analysis, the improvement in the mean PPD was significant in both groups when comparing the data before and after the intervention. Given that significant between-group differences were not found in the average PPD or number of sites with BOP, we could not conclude that advanced self-care was superior to standard care alone in this trial. Health behavior changes appeared to have a major role in the improvements in addition to periodontal treatment in the hospital or clinic. Gentle instructions and training with an adequate explanation given by oral care providers would probably result in health behavior changes and sincere efforts to continue self-care on a daily basis among almost all patients.

## Limitations

This trial had some limitations. The first limitation was the lack of blinding. Given the nature of the intervention and control, we could not mask care providers and subjects to treatments; therefore, our results may reflect expectation bias. Second, the assessment of adherence to advanced self-care depended on a self-reported individual diary. Although we considered 70% implementation as successful advanced periodontal self-care in this trial, the selection of the minimal care frequency should remain a consideration. Third, the 3-month follow-up period was chosen based on a previous study [12–15] and selected in view of the burden on subjects. A longer period, which enables further improvement in bacteremia, might be required to alter endothelial function in patients with mild-to-moderate periodontal diseases.

## Conclusion

Advanced periodontal self-care for three months did not significantly improve FMD as compared to standard care in patients with early-stage periodontal diseases. Neither care significantly improved FMD despite its effectiveness in improving the periodontal status.

## Supporting information

**S1 Checklist. CONSORT non-pharmacologic treatment extension checklist.**
(DOCX)

**S1 File. Trial protocol.**
(DOCX)

**S1 Fig. Custom-manufactured tray.**
(TIF)

**S2 Fig. Effect of periodontal care on vascular function in the intention-to-treat analysis.**
Box and whisker plot of FMD (A) and serum ADMA (B) levels by group. The box contains values between the 25th and 75th percentiles (central line, median). Vertical lines represent the minimum and maximum. P-values were calculated using the paired Student's $t$-test (FMD) or Wilcoxon's signed-rank test (serum ADMA level) for changes from baseline and the unpaired $t$-test (FMD) or the Mann–Whitney U test (serum ADMA level) for group differences. FMD, brachial artery dilatation; AMDA, asymmetric dimethylarginine.
(TIF)

**S3 Fig. Effect of periodontal care on vascular function in the per-protocol analysis.**
Box and whisker plot of FMD (A) and serum ADMA levels (B) by group. The box contains

values between the 25th and 75th percentiles (central line, median) of serum ADMA levels. Vertical lines represent the minimum and maximum. P-values were calculated using the paired Student's *t*-test (FMD) or Wilcoxon's signed-rank test (serum ADMA level) for changes from baseline and the unpaired *t*-test (FMD) or the Mann–Whitney U test (serum ADMA level) for group differences. FMD, brachial artery dilatation; AMDA, asymmetric dimethylarginine. (TIF)

**S1 Table. Assessment of vascular function in the per-protocol analysis.**
(DOCX)

**S2 Table. Periodontal status in the per-protocol analysis.**
(DOCX)

## Acknowledgments

The authors are grateful to the staff of the Ariyoshi Dental Clinic and the Kioicho Plaza Clinic for their invaluable help with the recruitment process. The authors would also like to thank Ms. Tomomi Kurashita, Ms. Eri Sato, Ms. Nozomi Aiba, Ms. Yurie Uno., Ms. Mitsuko Endo, and Ms. Juli Tomitani for their oral health care provision to the patients and Dr. Junko Kobayashi, Ms. Masumi Kawauchino, Ms. Chiemi Honda, and Ms. Hideko Aratani for their contribution in collecting data on serum ADMA levels.

## Author Contributions

**Conceptualization:** Ayako Okada, Takatoshi Murata, Khairul Matin, Nobuhiro Hanada.

**Data curation:** Ayako Okada, Meu Ariyoshi, Ryoko Otsuka, Mamiko Yamashita.

**Formal analysis:** Takatoshi Murata.

**Funding acquisition:** Ayako Okada, Takatoshi Murata, Khairul Matin, Meu Ariyoshi, Ryoko Otsuka, Tsutomu Sato, Nobuhiro Hanada.

**Investigation:** Ayako Okada, Takatoshi Murata, Khairul Matin, Meu Ariyoshi, Ryoko Otsuka, Mamiko Yamashita, Masayuki Suzuki, Rumi Wakiyama, Ken Tateno, Megumi Suzuki, Hitomi Aoyagi, Hiromi Uematsu, Akiko Imamura, Miki Kosaka, Tomoko Mizukaki, Tsutomu Sato, Hiroshi Kawahara, Nobuhiro Hanada.

**Methodology:** Ayako Okada, Takatoshi Murata, Khairul Matin, Nobuhiro Hanada.

**Project administration:** Ayako Okada, Meu Ariyoshi.

**Resources:** Meu Ariyoshi, Tsutomu Sato, Nobuhiro Hanada.

**Supervision:** Nobuhiro Hanada.

**Validation:** Nobuhiro Hanada.

**Visualization:** Takatoshi Murata.

**Writing – original draft:** Takatoshi Murata.

**Writing – review & editing:** Ayako Okada, Khairul Matin, Meu Ariyoshi, Ryoko Otsuka, Mamiko Yamashita, Masayuki Suzuki, Rumi Wakiyama, Ken Tateno, Megumi Suzuki, Hitomi Aoyagi, Hiromi Uematsu, Akiko Imamura, Miki Kosaka, Tomoko Mizukaki, Tsutomu Sato, Hiroshi Kawahara, Nobuhiro Hanada.

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
