## [Decision Letter · Decision Letter 0]

2 Jul 2021

PONE-D-21-11565

Effect of advanced periodontal self-care in patients with early-stage periodontal diseases on endothelial function: An open-label, randomized controlled trial

PLOS ONE

Dear Dr. Murata,

Thank you for submitting your manuscript to PLOS ONE. After careful consideration, we feel that it has merit but does not fully meet PLOS ONE’s publication criteria as it currently stands. Therefore, we invite you to submit a revised version of the manuscript that addresses the points raised during the review process.

Having intensively reviewed your draft, our external reviewers differed with their final recommendations, and, thus, I have double checked your revised version (at least to some extent), to come to a more balanced decision (see R #1). All in all, our reviewers have identified shortcomings considered reasonable with regard to both PLOS ONE’s quality standards and our readership's expectations. Therefore, we invite you to submit a thoroughly and completely revised version of the manuscript that addresses EACH AND EVERY point raised during the current review process. Please note that a non-convincing revision (not considered acceptable with regard to language, content, reviewers' constructive criticism, generalizable conclusions, and/or Authors' Guidelines) must lead to outright reject. 

We look forward to receiving your revised manuscript.

Kind regards,

Andrej M Kielbassa

Academic Editor

PLOS ONE

Journal Requirements:

[Dr. Khairul Matin and Dr. Nobuhiro Hanada received research grant from Medoc International Co. Ltd. Dr. Nobuhiro Hanada received a research grant from Shiken Corp.

Medoc International Co. Ltd.: http://www.medoc.co.jp/company

Shiken Corp.: https://www.shiken-jp.com].

Reviewers' comments:

Reviewer's Responses to Questions

**Comments to the Author**

1. Is the manuscript technically sound, and do the data support the conclusions?

Reviewer #1: Partly

Reviewer #2: Partly

Reviewer #3: Yes

2. Has the statistical analysis been performed appropriately and rigorously? 

Reviewer #1: Yes

Reviewer #2: Yes

Reviewer #3: No

3. Have the authors made all data underlying the findings in their manuscript fully available?

Reviewer #1: Yes

Reviewer #2: Yes

Reviewer #3: Yes

4. Is the manuscript presented in an intelligible fashion and written in standard English?

Reviewer #1: Yes

Reviewer #2: Yes

Reviewer #3: Yes

5. Review Comments to the Author

Reviewer #1: Abstract

- Remember to provide as much as information as possible here. Plos One accepts 300 words with this section, and you should enrich your information.

- With your conclusions in mind, please revise carefully, and see comments given below.

Intro

- Aims and objectives have been satisfyingly elaborated.

- Please note that this section must not provide what you have done (but, instead, what you were going to do). Phrases like "In an RCT, we determined the effect (...)." or "We attempted to advance (...)." must be carefully revised.

- Please provide a reasonable null hypothesis. Remember that H0 must be deducible from the foregoing thoughts.

Meths

- What is meant when referring to "whole-mouth scaling and root planning"? Performed in one visit? Please clarify.

- Please refer to the CONSORT statement with your full text.

- Additionally, please stick to the SQUIRE guidelines. You are reporting new knowledge about how to improve healthcare, right?

- This was an intention-to-treat study, right? Please refer to the respective analysis.

- "ultrasonic scaler", "hand instruments", "dental plaque-disclosing agent", "polishing paste", "rotating rubber cup", "rotating brush", and so on: Please note with ALL materials (including chemicals) and methodologies (including statistical software), please use general names with your text, followed by (brand name; manufacturer, city, STATE (abbreviated, if US), country) in parentheses. Stick to semicolon. Revise thoroughly throughout your text.

- Remember that reproducibility is the cornerstone of scientific advancement. Outputs like exact methodology protocols empower researchers to go one step further in contextualizing their work to ensure it remains replicable. This section has not been satisfying elaborated.

- Do not use legal terms with your text. Delete "Corporation", "Corp.", "Co. Ltd.", "Co.", "Ltd.", "Inc.",

- "A single examiner collected (...)." Please provide initials in parentheses.

- "−30°C" must read "-30 °C". Use minus/hyphen instead of dash. The unit is "°C", and must be separated from the number.

- Report exact p-values for all values greater than or equal to 0.001 (note the 3-digit basis). P-values less than 0.001 may be expressed as p < 0.001. Remember to use lowercase letter p. See Authors' Guidelines, and revise thoroughly.

Results

- Again, double check and revise p values.

Disc

- Stick to H0 when staring this section. Remember that H0 can be rejected or not rejected.

- This section would seem perfectible, please add more discursive thoughts on your outcome.

Concl

- You aimed to "determine the effect of advanced periodontal self-care in patients with early-stage periodontal diseases on endothelial function". Additionally, you "attempted to advance periodontal self-care in patients with early-stage periodontal disease to determine any effect on ACVD-related vascular function markers FMD and ADMA." Hence, please adapt your conclusion ("Advanced periodontal self-care, in addition to standard care, did not result in better vascular function in patients with mild-to-moderate periodontal diseases in this trial.") to your aims. Do not simply repeat your results, but provide a reasonable extension of your outcome.

In total, this submitted draft would seem interesting, is considered easily intelligible, and should be worth following after revision. This paper is ready for external review.

Reviewer #2: In the manuscript entitled: “Effect of advanced periodontal self-care in patients with early-stage periodontal diseases on endothelial function: An open-label, randomized controlled trial”, the authors identified the effects of advanced periodontal selfcare on endothelial function in patients with early-stage periodontal disease. The study was designed as a parallel group, 3-month follow-up, open-label, randomized controlled trial for evaluating the effectiveness of periodontal care against atherosclerotic cardiovascular disease.

The authors found that both groups demonstrated improved periodontal status. No significant improvements in FMD were observed in the control or test group. No significant changes in serum ADMA levels were observed. No significant between-group differences were found in FMD or serum ADMA levels.

The authors concluded that periodontal care for a 3-month duration did not provide better endothelial function in patients with early-stage periodontal diseases.

Major comments:

In general, the idea and innovation of this study, regards analysis of eriodontal self-care in patients with early-stage periodontal diseases on endothelial function is interesting, because the role of these factors in dentistry are validated but further studies on this topic could be an innovative issue in this field could be open a creative matter of debate in literature by adding new information. Moreover, there are few reports in the literature that studied this interesting topic with this kind of study design.

The study was well conducted by the authors; However, there are some concerns to revise that are described below.

The introduction section resumes the existing knowledge regarding the important factor linked with periodontal inflammation.

However, as the importance of the topic, the reviewer strongly recommends, before a further re-evaluation of the manuscript, to update the literature through read, discuss and must cites in the references with great attention all of those recent interesting articles, that helps the authors to better introduce and discuss the role of periodontitis and related biomarkers as cause of implant failure (Galectin, NLRP3): 1) Isola G, Polizzi A, Alibrandi A, Williams RC, Lo Giudice A. Analysis of galectin-3 levels as a source of coronary heart disease risk during periodontitis. J Periodontal Res. 2021 Jun;56(3):597-605. doi: 10.1111/jre.12860. 2) Isola G, Polizzi A, Santonocito S, Alibrandi A, Williams RC. Periodontitis activates the NLRP3 inflammasome in serum and saliva. J Periodontol. 2021 May 19. doi: 10.1002/JPER.21-0049. 3) Isola G, Lo Giudice A, Polizzi A, Alibrandi A, Murabito P, Indelicato F. Identification of the different salivary Interleukin-6 profiles in patients with periodontitis: A cross-sectional study. Arch Oral Biol. 2021 Feb;122:104997. doi: 10.1016/j.archoralbio.2020.104997.

The authors should be better specified, at the end of the introduction section, the rational of the study and the aim of the study. In the material and methods section, should better clarify randomization and periodontal examination. Moreover, please more specifiy the clinicians involved in the different stages of the stusy.

The discussion section appears well organized with the relevant paper that support the conclusions, even if the authors should better discuss the relationship between periodontitis and endotelial dysfunction. The conclusion should reinforce in light of the discussions.

In conclusion, I am sure that the authors are fine clinicians who achieve very nice results with their adopted protocol. However, this study, in my view does not in its current form satisfy a very high scientific requirement for publication in this journal and requests a revision before a futher re-evaluation of the manuscript.

Minor Comments:

Abstract:

- Better formulate the abstract section by better describing the aim of the study

Introduction:

- Please refer to major comments

Discussion

- Please add a specific sentence that clarifies the results obtained in the first part of the discussion

- Page 16 last paragraph: Please reorganize this paragraph that is not clear

Reviewer #3: 1. Sample size calculation needs more details. What is the mean (SD) of each group at each time point? What test was used?

2. Linear mixed models (repeated ANOVA) may be considered for analyzing the outcomes. If data/model residuals follow normal distributions, it is more powerful to use parametric methods.

3. Results:

a. Table 1: add p values. Specify methods used for comparison in the footnote.

b. Periodontal condition: move this section after “Vascular function” as they are secondary outcomes. Show median (IQR) for each group at each time point, not just improvement. Add all p values for within-group comparisons and between-group comparisons.

c. Vascular function: add a table with detailed statistics, e.g. mean (SD) /median (IQR) for each group at each time point. Add all p values for within-group comparisons and between-group comparisons. Need to perform a consistency analysis for ITT and per-protocol also.

d. Figure 2: using boxplot for both outcomes. Specify clearly which methods were used for generating the p values.

e. Line 275 is not a complete sentence.

Consort checklist:

1) Outcomes are clearly defined.

2) Sample size needs more details for mean (SD) in each group, not mean difference (SD). What test was used?

3) Method used to generate random allocation sequence was not mentioned.

4) Allocation concealment was clearly described.

5) Blinding is not available.

6) Details of outcomes and estimation were not provided. Need means (SD) for each group at each time point and mean different (with 95% CI). A table is necessary. Effect size was not provided either.

7) Important harms or unintended effects were not discussed.

8) The registry and the registration number were reported.

9) The trial protocol is attached.

10) Sources of funding was described.

6. PLOS authors have the option to publish the peer review history of their article (what does this mean?). If published, this will include your full peer review and any attached files.

Reviewer #1: No

Reviewer #2: No

Reviewer #3: No

---

## [Author Response · Author response to Decision Letter 0]

22 Jul 2021

Response to Reviewers

We are grateful to each reviewer for the valuable feedback and critical comments, which have helped us improve our manuscript considerably. As indicated in the responses that follow, we have taken all these comments and suggestions into account in the revised version of our manuscript.

Point-by-point responses to Reviewer #1

Abstract

Comment:

- Remember to provide as much as information as possible here. Plos One accepts 300 words with this section, and you should enrich your information.

- With your conclusions in mind, please revise carefully, and see comments given below.

Response:

Based on this comment, we have added the sentences of the background and aim of the study in the revised manuscript (Lines 33–41). The word count of the Abstract is now 295 words.

We have taken all the Reviewer #1's comments and suggestions into account in the revised version of our manuscript.

Intro

Comment:

- Aims and objectives have been satisfyingly elaborated.

Response:

Thank you for your positive evaluation.

Comment:

- Please note that this section must not provide what you have done (but, instead, what you were going to do). Phrases like "In an RCT, we determined the effect (...)." or "We attempted to advance (...)." must be carefully revised.

Response:

Based on this comment, we have replaced “Therefore, advanced self-care may contribute to improving FMD. In an RCT, we determined the effect of advanced periodontal self-care in patients with early-stage periodontal diseases on endothelial function. FMD was set as the primary outcome, while serum asymmetric dimethylarginine (ADMA; an endogenous NO synthase inhibitor) level was the secondary outcome. Elevated serum ADMA is thought to impair endothelial function and, thus, promote atherosclerosis [20]. We attempted to advance periodontal self-care in patients with early-stage periodontal disease to determine any effect on ACVD-related vascular function markers FMD and ADMA.” in the original manuscript (lines 82–90) with “Therefore, we hypothesized that advanced self-care improves endothelial function. This randomized clinical trial aimed to investigate the effect of advanced self-care on ACVD-related vascular function markers.” in the revised manuscript (Lines 89–91).

Comment:

- Please provide a reasonable null hypothesis. Remember that H0 must be deducible from the foregoing thoughts.

Response:

Based on this comment, we have added the following sentence:

“Our null hypothesis was that there was no difference in FMD or serum ADMA level between advanced self-care was compared and standard care in patients with early-stage periodontal disease” (Lines 91–92).

Meths

Comment:

- What is meant when referring to "whole-mouth scaling and root planning"? Performed in one visit? Please clarify.

Response:

We would like to mean “whole-mouth scaling in a single visit”. Based on this comment, we have replaced “whole-mouth scaling” in the original manuscript (Line 121) with “same-day full-mouth scaling” in the revised manuscript (Line 123).

The “same-day full-mouth root planning” was used in the title of a clinical trial report*.

We have deleted “root planning”. No subjects needed to receive root planning because of their low severity of periodontal diseases.

* Apatzidou DA and Kinane DF. Quadrant root planing versus same-day full-mouth root planing. I. Clinical findings. J Clin Periodontol. 2004; 31: 132-140. doi: 10.1111/j.0303-6979.2004.00461.x.

Comment:

- Please refer to the CONSORT statement with your full text.

Response:

We have reviewed overall manuscript to adjust it according to the CONSORT statement and replaced with “S1 CONSORT NPT Extension Checklist”.

Comment:

- Additionally, please stick to the SQUIRE guidelines. You are reporting new knowledge about how to improve healthcare, right?

Response:

Thank you for your advice. However, we are not focusing the improvement of the quality of healthcare. The effectiveness of oral care has not been established on the improvement of endothelial function. Therefore, we must clarify the effectiveness of oral care prior to the evaluation of quality improvement in this study. The reason for setting the standard care group as the control group is due to ethical decision. Given that all the subjects were diagnosed with early-stage periodontal diseases, they have a right to receive standard care. We would like you to understand that we are not reporting the improvement of the quality of healthcare.

Comment:

- This was an intention-to-treat study, right? Please refer to the respective analysis.

Response:

Based on this comment, we have added revised S1 Table, S2 Table, and S3 Figure showing per-protocol analysis on vascular function and periodontal status in the revised manuscript, respectively.

Comment:

- "ultrasonic scaler", "hand instruments", "dental plaque-disclosing agent", "polishing paste", "rotating rubber cup", "rotating brush", and so on: Please note with ALL materials (including chemicals) and methodologies (including statistical software), please use general names with your text, followed by (brand name; manufacturer, city, STATE (abbreviated, if US), country) in parentheses. Stick to semicolon. Revise thoroughly throughout your text.

- Remember that reproducibility is the cornerstone of scientific advancement. Outputs like exact methodology protocols empower researchers to go one step further in contextualizing their work to ensure it remains replicable. This section has not been satisfying elaborated.

Response:

Based on this comment, we have reviewed overall the manuscript and added the required information about the equipment.

“ultrasonic scaler (Varios970; Nakanishi, Tochigi, Japan)” (Line 129).

“hand instruments (FP scaler; Feed, Yokohama, Japan)” (Line 129).

“dental floss （Reach No-waxed; Johnson & Johnson, Tokyo, Japan)” (Line 130)

“dental plaque-disclosing agent (Merssage PC Pellet Blue; Shofu, Kyoto, Japan)” (Line 131).

“polishing paste (PTC Paste Fine/PTC Paste Regular; GC, Tokyo, Japan)” (Line 133)

“rotating rubber cup (FP rubber; Feed, Yokohama, Japan)” (Lines 133–134).

“rotating brush (FP profy brush; Feed, Yokohama, Japan)” (Line 134).

“dental plaster (New Plastone II; GC, Tokyo, Japan)” (Line 158).

“vacuum-forming machine (Biostar; Scheu Dental, Iserlohn, Germany:)” (Lines 161–162).

Comment:

- Do not use legal terms with your text. Delete "Corporation", "Corp.", "Co. Ltd.", "Co.", "Ltd.", "Inc.",

Response:

Based on this comment, we have gone through the entire manuscript and deleted the legal terms.

Comment:

- "A single examiner collected (...)." Please provide initials in parentheses.

Response:

Based on this comment, we have added the “(A.O.)” in the revised manuscript (Line 195).

Comment:

- "−30°C" must read "-30 °C". Use minus/hyphen instead of dash. The unit is "°C", and must be separated from the number.

Response:

Based on this comment, we have replaced “−30°C” in the original manuscript (Line 201) with “-30 ℃” in the revised manuscript (Line 218).

Comment:

- Report exact p-values for all values greater than or equal to 0.001 (note the 3-digit basis). P-values less than 0.001 may be expressed as p < 0.001. Remember to use lowercase letter p. See Authors' Guidelines, and revise thoroughly.

Response:

Based on this comment, we have reviewed overall the manuscript and revised the appropriate p-values.

Results

Comment:

- Again, double check and revise p values.

Response:

Based on this comment, we have reviewed overall the manuscript and revised the appropriate p-values.

Disc

Comment:

- Stick to H0 when staring this section. Remember that H0 can be rejected or not rejected.

Response:

Based on this comment, we have added the following sentence:

Advanced periodontal self-care for three months did not significantly improve FMD as compared to standard care (Lines 303–304).

Comment:

- This section would seem perfectible, please add more discursive thoughts on your outcome.

Response:

Thank you for your positive evaluation. Based on this comment, we have added the following sentences in the revised manuscript:

Reduced FMD is associated with the risk of ACVD and improves with risk-reduction therapy. Consequently, endothelial function has been defined as an “excellent barometer” of vascular health. FMD assessing vascular endothelial function might be ideal for exploring factors associated with the improvement of vascular health (Lines 308–311).

Concl

Comment:

- You aimed to "determine the effect of advanced periodontal self-care in patients with early-stage periodontal diseases on endothelial function". Additionally, you "attempted to advance periodontal self-care in patients with early-stage periodontal disease to determine any effect on ACVD-related vascular function markers FMD and ADMA." Hence, please adapt your conclusion ("Advanced periodontal self-care, in addition to standard care, did not result in better vascular function in patients with mild-to-moderate periodontal diseases in this trial.") to your aims. Do not simply repeat your results, but provide a reasonable extension of your outcome.

Response:

Based on this comment, we have replaced “Advanced periodontal self-care, in addition to standard care, did not result in better vascular function in patients with mild-to-moderate periodontal diseases in this trial” in the original manuscript (Lines 329–330) with “Advanced periodontal self-care for three months did not significantly improve FMD as compared to standard care in patients with early-stage periodontal diseases. Neither care significantly improved FMD despite its effectiveness in improving periodontal status. (Lines 362–364).

Comment:

In total, this submitted draft would seem interesting, is considered easily intelligible, and should be worth following after revision. This paper is ready for external review.

Response:

Thank you for your positive evaluation. We hope that our explanations and revisions are satisfactory.

 

Point-by-point responses to Reviewer #2

In the manuscript entitled: “Effect of advanced periodontal self-care in patients with early-stage periodontal diseases on endothelial function: An open-label, randomized controlled trial”, the authors identified the effects of advanced periodontal selfcare on endothelial function in patients with early-stage periodontal disease. The study was designed as a parallel group, 3-month follow-up, open-label, randomized controlled trial for evaluating the effectiveness of periodontal care against atherosclerotic cardiovascular disease.

The authors found that both groups demonstrated improved periodontal status. No significant improvements in FMD were observed in the control or test group. No significant changes in serum ADMA levels were observed. No significant between-group differences were found in FMD or serum ADMA levels.

The authors concluded that periodontal care for a 3-month duration did not provide better endothelial function in patients with early-stage periodontal diseases.

Major comments:

In general, the idea and innovation of this study, regards analysis of eriodontal self-care in patients with early-stage periodontal diseases on endothelial function is interesting, because the role of these factors in dentistry are validated but further studies on this topic could be an innovative issue in this field could be open a creative matter of debate in literature by adding new information. Moreover, there are few reports in the literature that studied this interesting topic with this kind of study design.

The study was well conducted by the authors; However, there are some concerns to revise that are described below.

Comment:

The introduction section resumes the existing knowledge regarding the important factor linked with periodontal inflammation. However, as the importance of the topic, the reviewer strongly recommends, before a further re-evaluation of the manuscript, to update the literature through read, discuss and must cites in the references with great attention all of those recent interesting articles, that helps the authors to better introduce and discuss the role of periodontitis and related biomarkers as cause of implant failure (Galectin, NLRP3): 1) Isola G, Polizzi A, Alibrandi A, Williams RC, Lo Giudice A. Analysis of galectin-3 levels as a source of coronary heart disease risk during periodontitis. J Periodontal Res. 2021 Jun;56(3):597-605. doi: 10.1111/jre.12860. 2) Isola G, Polizzi A, Santonocito S, Alibrandi A, Williams RC. Periodontitis activates the NLRP3 inflammasome in serum and saliva. J Periodontol. 2021 May 19. doi: 10.1002/JPER.21-0049. 3) Isola G, Lo Giudice A, Polizzi A, Alibrandi A, Murabito P, Indelicato F. Identification of the different salivary Interleukin-6 profiles in patients with periodontitis: A cross-sectional study. Arch Oral Biol. 2021 Feb;122:104997. doi: 10.1016/j.archoralbio.2020.104997.

Response:

Thank you for your critical advice. As mentioned by Reviewer #2, periodontal inflammation may induce atherosclerotic cardiovascular disease through not only bacteremia but also signal transduction involved in inflammatory mediators.

Based on this comment, we have added the following sentence with citations of all the introduced references in the revised manuscript:

“Furthermore, it has also been suggested that inflammatory mediators attributable to periodontal diseases are associated with ACVD” (Lines 61 –62).

Comment:

The authors should be better specified, at the end of the introduction section, the rational of the study and the aim of the study.

Response:

Based on this comment, we have added the following sentence in the revised manuscript:

Therefore, we hypothesized that advanced self-care improves endothelial function. The aim of this randomized clinical trial was to investigate the effect of advanced self-care on ACVD-related vascular function markers.” (Lines 89–91).

Comment:

In the material and methods section, should better clarify randomization* and periodontal examination**.

Response:

*Based on this comment, we have added the following sentences in the revised manuscript:

“All the researchers confirmed that no one could identify the number” (Lines 184–185).

“After eligibility assessment, each patient selected as a study subject made an appointment for random allocation at their convenience.” (Lines 188–189).

“Balloting was done on a first-come, first-served basis. The allocated Arabic number was particular for each subject, and the number was never used again” (Lines 190–192).

** Based on this comment, we have replaced “A single examiner collected the following clinical data from six sites around each tooth” in the original manuscript (Line 182) with “A single examiner (A.O.) collected the following clinical data from six sites (mesio-buccal, mid-buccal, disto-buccal, mesio-lingual, mid- lingual, and disto-lingual) around each tooth using a color-coded periodontal probe (PO-9; Nippon Shiken, Tokyo, Japan) in the revised manuscript (Lines 195–197).

Comment:

Moreover, please more specific the clinicians involved in the different stages of the study.

Response:

Based on this comment, we have provided initials of each clinician or qualified medical personnel in parentheses in each of the different stages in the revised manuscript:

Authors: R.O., M.S., H.A., H.U., A.I., Non-authors: refer to Acknowledgments; T.K., E.S., N.A., Y.U., M.E., J.T. (Lines 125–126),

T.M. (Line 136, 146)

A.O. (Line 195, 214).

Comment:

The discussion section appears well organized with the relevant paper that support the conclusions, even if the authors should better discuss the relationship between periodontitis and endothelial dysfunction. 

Response:

Based on this comment, we have added the following sentence with a citation concerning ACVD through inflammatory mediators in the revised manuscript:

“In addition, production of inflammatory mediators attributable to periodontal diseases might be associated with ACVD (Lines 306–308).

Comment:

The conclusion should reinforce in light of the discussions.

Response:

Based on this comment, we have replaced “Advanced periodontal self-care, in addition to standard care, did not result in better vascular function in patients with mild-to-moderate periodontal diseases in this trial” in the original manuscript (Lines 329–330) with “Advanced periodontal self-care for three months did not significantly improve FMD as compared to standard care in patients with early-stage periodontal diseases. Neither care significantly improved FMD despite its effectiveness in improving the periodontal status.” in the revised manuscript (Lines 362–364).

Comment:

In conclusion, I am sure that the authors are fine clinicians who achieve very nice results with their adopted protocol. However, this study, in my view does not in its current form satisfy a very high scientific requirement for publication in this journal and requests a revision before a further re-evaluation of the manuscript.

Response:

We have addressed all the comments by Reviewer #2. We hope that our explanations and revisions are satisfactory.

Minor Comments

Abstract

Comment:

- Better formulate the Abstract section by better describing the aim of the study

Response:

Based on this comment, we have added the following sentence in the Abstract:

This randomized clinical trial aimed to investigate the effect of advanced self-care on ACVD-related vascular function markers flow-mediated brachial artery dilatation (FMD) and serum asymmetric dimethylarginine (ADMA) levels in patients with early-stage periodontal disease (lines 37–41)

Introduction

Comment:

- Please refer to major comments

Response:

Based on the major comment above, we have added the following sentence with citation in the revised manuscript:

Furthermore, it has also been suggested that inflammatory mediators attributable to periodontal diseases are associated with ACVD (Lines 61–62).

Discussion

Comment:

- Please add a specific sentence that clarifies the results obtained in the first part of the discussion

Response:

Based on this comment, we have added the following sentences in the revised manuscript:

Advanced periodontal self-care for three months did not significantly improve FMD as compared to standard care. (Lines 303–304).

Comment:

- Page 16 last paragraph: Please reorganize this paragraph that is not clear

Response:

Based on this comment, we have added “than in patients with early-stage periodontal diseases” in the revised manuscript (Line 324).

 

Point-by-point responses to Reviewer #3

Comment:

1. Sample size calculation needs more details. What is the mean (SD) of each group at each time point? What test was used?

Response:

We did not have enough data prior to the trial. Therefore, we referred to previous similar trials (Tonetti MS, et al. Treatment of periodontitis and endothelial function. N Engl J Med. 2007; 356: 911-920.) as described in the original manuscript (Lines 164–167) and the revised manuscript (Lines 173–176). The authors set: effect size; 1%, standard deviation of the mean difference: 1.67%.

We have added the following sentence “We employed a priori power analysis for sample size calculation” in the revised manuscript (Line 173).

Comment:

2. Linear mixed models (repeated ANOVA) may be considered for analyzing the outcomes. If data/model residuals follow normal distributions, it is more powerful to use parametric methods.

Response:

Thank you for your advice. There are two time points in this trial (baseline and endpoint). Therefore, the paired t-test is used for comparison between baseline and endpoint. Repeated ANOVA should be used in cases where there are more than three time points.

Comment:

3. Results:

a. Table 1: add p values. Specify methods used for comparison in the footnote.

Response:

We show the baseline comparison in Table 1. As described in “How to Report Statistics in Medicine (2nd ed.) (Thomas A. Lang, Michelle Secic. American College of Physicians Philadelphia 2006, pp 207-2081), the p-values for baseline comparisons would not be necessary in randomized trials. Therefore, we did not report the p-values for baseline comparison. We have removed the description of the baseline comparison on vascular function from the original manuscript (Lines 256–257).

1 It is not necessary to report the p-values for baseline comparisons in randomized trials. In such trials, any differences between groups in baseline variables will be the result of chance because participants were assigned to groups at random. Baseline comparisons do need to be made, however, to identify any statistical imbalances that may need to be adjusted for in the final multivariable model. If p-values are reported for baseline comparisons in a randomized trial, they should be interpreted only as measure of the strength of the imbalance between the groups, not as evidence of bias.

Comment:

b. Periodontal condition: move this section after “Vascular function” as they are secondary outcomes*. Show median (IQR) for each group at each time point, not just improvement**. Add all p values for within-group comparisons and between-group comparisons**.

Response:

* Based on this comment, the description of “Vascular function” is followed by that of “Periodontal condition” in the revised manuscript.

** Based on these comments, we have replaced Table 2 and S1 Table in the original manuscript with revised Table 3 (ITT) and S2 Table (per-protocol analysis) in the revised manuscript, respectively. The revised tables show the median (IQR) for each group at endpoint with p-values in addition to the mean improvement (95% CI). Baseline data have already been described in Table 1. The revised S2 Table includes baseline data because of no descriptions on baseline for per-protocol analysis in the manuscript. We have removed statistical results from the text to avoid duplicate descriptions.

Comment:

c. Vascular function: add a table with detailed statistics, e.g. mean (SD) /median (IQR) for each group at each time point. Add all p values for within-group comparisons and between-group comparisons. Need to perform a consistency analysis for ITT and per-protocol also.

Response:

Based on this comment, we have added revised Table 2 (ITT) and S1 Table (per-protocol analysis) in the revised manuscript. The revised tables show the mean (SD) or median (IQR) for each group at endpoint with p-values in addition to the mean improvement (95% CI). Baseline data have already been described in Table 1. The revised S1 Table includes baseline data because of the absence of descriptions on baseline for per-protocol analysis in the manuscript. We have removed statistical results from the text except mean improvement between group to avoid duplicate descriptions. We have deleted Fig. 2 in the original manuscript to avoid duplicate descriptions. However, figures are generally preferred to tables for presentations of comparisons2. Therefore, we have shown the revised figures as supporting information (S2 Fig and S3 Fig).

2 How to Report Statistics in Medicine (2nd ed.). Thomas A. Lang, Michelle Secic. American College of Physicians Philadelphia 2006, pp 328

“We recommend that every effort be made to present comparisons in figures rather than in tables, even when the amount of data is small.”

Comment:

d. Figure 2: using boxplot for both outcomes. Specify clearly which methods were used for generating the p values.

Response:

Based on this comment, we have replaced the original Figure 2 with revised S2 Fig, indicating the results with boxplots. The boxplots are presented as supporting information to avoid duplicate descriptions. We have also adjusted the corresponding captions. Statistical procedures were described in the captions.

Comment:

e. Line 275 is not a complete sentence.

Response:

Line 275 contains definitions of abbreviations and a part of Figure captions. We have undone the line break in the revised manuscript.

CONSORT checklist:

1) Outcomes are clearly defined.

Complete

2) Sample size needs more details for mean (SD) in each group, not mean difference (SD). What test was used?

Response:

As described above, we did not have enough data prior trial. Therefore, we referred to previous similar trials (Tonetti MS, et al. Treatment of periodontitis and endothelial function. N Engl J Med. 2007; 356: 911-920.) as described in the original (Lines 164–167) and revised (Lines 173–176) manuscripts. The authors set: effect size; 1%, standard deviation of the mean difference: 1.67%.

We have added the following sentence in the revised manuscript “We used a priori power analysis for sample size calculation” (Line 173).

3) Method used to generate random allocation sequence was not mentioned.

Response:

We have added the following sentences to clarify the random allocation sequence in the revised manuscript:

“All the researchers confirmed that no one could identify the number (Lines 184–185).

“After eligibility assessment, each patient selected as a study subject made an appointment for random allocation at their convenience.” (Lines 188–189).

“Balloting was done on a first-come, first-served basis. The allocated Arabic number was particular for each subject, and the number was never used again.” (Lines 190–192).

4) Allocation concealment was clearly described.

Complete

5) Blinding is not available.

Response:

This trial is open-labeled.

6) Details of outcomes and estimation were not provided*. Need means (SD) for each group at each time point and mean different (with 95% CI)**. A table is necessary**. Effect size was not provided either*.

Response:

* We have added the following sentences to the revised manuscript:

“An improvement in FMD of at least 1% difference was not achieved between the groups.” (Line 261).

We set the effect size at 1% difference in FMD as described in the original (Line 165) and revised (Line 174) manuscripts.

** We have added revised Table 2 (ITT) and S1 Table (per-protocol analysis) showing the mean ± SD/median (IQR) for each group at endpoint and mean difference (with 95% CI) in the revised manuscript.

7) Important harms or unintended effects were not discussed.

Response:

We have added the following sentence:

Harms

No study-related serious adverse events occurred in any of the study participants. Nonsurgical periodontal treatment is basically low-risk. None of the participants required any dental therapy during the study. (Lines 297–300).

8) The registry and the registration number were reported.

Complete

9) The trial protocol is attached.

Complete

10) Sources of funding was described.

Complete

---

## [Decision Letter · Decision Letter 1]

27 Jul 2021

PONE-D-21-11565R1

Effect of advanced periodontal self-care in patients with early-stage periodontal diseases on endothelial function: An open-label, randomized controlled trial

PLOS ONE

Dear Dr. Murata,

Thank you for submitting your manuscript to PLOS ONE. After careful consideration, we feel that it has merit but does not fully meet PLOS ONE’s publication criteria as it currently stands. Therefore, we invite you to submit a revised version of the manuscript that addresses the points raised during the review process.

Having intensively reviewed your revised draft, our external reviewers still differed with their final recommendations, at least to some extent. I have again double checked your revised version, to come to a more balanced decision (see R #1). All in all, our  identified shortcomings are considered reasonable with regard to both PLOS ONE’s quality standards and our readership's expectations. Therefore, we invite you to submit a carefully revised version of the manuscript that addresses EACH AND EVERY point raised during the current review process. Please note that a further non-convincing revision (not considered acceptable with regard to language, content, reviewers' constructive criticism, generalizable conclusions, and/or Authors' Guidelines) must lead to outright reject. 

We look forward to receiving your revised manuscript.

Kind regards,

Andrej M Kielbassa

Academic Editor

PLOS ONE

Reviewers' comments:

Reviewer's Responses to Questions

**Comments to the Author**

1. If the authors have adequately addressed your comments raised in a previous round of review and you feel that this manuscript is now acceptable for publication, you may indicate that here to bypass the “Comments to the Author” section, enter your conflict of interest statement in the “Confidential to Editor” section, and submit your "Accept" recommendation.

Reviewer #1: (No Response)

Reviewer #2: All comments have been addressed

Reviewer #3: All comments have been addressed

2. Is the manuscript technically sound, and do the data support the conclusions?

Reviewer #1: No

Reviewer #2: Yes

Reviewer #3: (No Response)

3. Has the statistical analysis been performed appropriately and rigorously? 

Reviewer #1: Yes

Reviewer #2: Yes

Reviewer #3: (No Response)

4. Have the authors made all data underlying the findings in their manuscript fully available?

Reviewer #1: No

Reviewer #2: Yes

Reviewer #3: (No Response)

5. Is the manuscript presented in an intelligible fashion and written in standard English?

Reviewer #1: Yes

Reviewer #2: Yes

Reviewer #3: (No Response)

6. Review Comments to the Author

Reviewer #1: No doubt, this revised draft has been considerably increased. However, some minor aspects would still seem in need of further revisions, please see below.

- With reference to your Abstract section, do not provide a shortened literature review, please. Instead, please focus more intensely on your outcome.

- "The immune response following persistent bacteremia from periodontal lesions may lead to ACVD [3]. Furthermore, it has also been suggested that inflammatory mediators attributable to periodontal diseases are associated with ACVD [4-6]." Consequently, "bacteremia-induced ACVD routinely occurs even in the early stage of periodontal disease [2,19]". Undoubtedly, this would seem right. However, this would lead to the idea of a thorough eradication of all biofilms deemed responsible for infection, persistent bacteremia, and presence of inflammatory mediators (due to the biofilm). This refers to your "same-day full-mouth scaling, an ultrasonic scaler, along with hand instruments, and dental floss, and polishing paste with rotating rubber cup/brush", which is not deemed sufficient (even if widely established). Please refer to https://pubmed.ncbi.nlm.nih.gov/34269042/; there you will see that "(not only supragingival) biofilm removal by means of air polishing combined with low-abrasive erythritol has been shown to be more efficacious than the traditional polishing method". Thus, your applied methodology should be discussed more thoroughly, in particular since "thorough biofilm removal clearly results in delayed plaque regrowth rates", thus potentially showing a clear influence on inflammatory mediators and ACVD, in particular in combination with your "extra intervention in addition to standard self-care". Again, go to https://doi.org/10.3290/j.qi.b1763661, and discuss, both with regard to general effects and with regard to possible limitations.

- "We considered a 70% implementation rate as successful advanced periodontal self-care." Do you mean "successfully advanced periodontal self-care"?

- "(1–29 for smokers and 1–81 for nonsmokers)" should read "(1 to 29 for smokers and 1 to 81 for nonsmokers)".

- "All the researchers confirmed that no one could identify the number." This would not seem clear, please clarify.

- "(Unexef18G, UNEX, Nagoya, Japan)" must read "(Unexef18G; UNEX, Nagoya, Japan)". Double check thoroughly.

- With your results section, please add exact p values. (See, for example, "Significant improvements in the average PPD were observed in the control and test groups. The BOP score in the test group was significantly reduced, while that in the control group was reduced, although not significantly.").

- Again, with the first paragraph of your Disc section, please refer explicitly to H0, to provide a clear answer. "The findings were similar in the intention-to-treat and the per-protocol analysis." should read ""The findings were similar in the intention-to-treat and the per-protocol analysis; thus H0 was not rejected."

- "ADMA is a naturally occurring endogenous inhibitor of NO synthase." Reference(s) missing.

- Same with "It reduces NO production; consequently, it can lead to endothelial dysfunction and cardiovascular events." Please revise carefully, and double check your whole draft, to support statements.

- "Further investigations may be required in healthy subjects." To study what kind of effects? Please clarify.

- "A longer period might be required to alter endothelial function in patients with mild-to-moderate periodontal diseases." Please provide reasons why you think that prolonged follow-up periods would change the outcome.

- Regarding the reference list, again, revise for uniform formatting (see Guidelines). Format must be "Bürgers R, Eidt A, Frankenberger R, Rosentritt M, Schweikl H, Handel G, et al. The anti-adherence activity and bactericidal effect of microparticulate silver additives in composite resin materials. Arch Oral Biol. 2009; 54(6): 595-601. https://doi.org/10.1016/j.archoralbio.2009.03.004 PMID: 19375069". Please revise for spacebar use to separate year and volume. Provide issue numbers (if available). Revise for correct doi numbers, and provide PMID numbers.

Again, this study would seem worth following after revision. Compliments to the authors!

Reviewer #2: The authors have well addressed to all reviewer's comments. There are no futher issues in the present version of the manuscript.

Reviewer #3: (No Response)

7. PLOS authors have the option to publish the peer review history of their article (what does this mean?). If published, this will include your full peer review and any attached files.

Reviewer #1: No

Reviewer #2: No

Reviewer #3: No

---

## [Author Response · Author response to Decision Letter 1]

4 Aug 2021

Response to Reviewers

Point-by-point responses to Reviewer #1

Comment:

No doubt, this revised draft has been considerably increased. However, some minor aspects would still seem in need of further revisions, please see below.

Response:

We are grateful to Reviewer #1 for the second review and for providing advice. As indicated in the responses that follow, we have taken all these comments and suggestions into account in the revised version of our manuscript.

Comment:

- With reference to your Abstract section, do not provide a shortened literature review, please. Instead, please focus more intensely on your outcome.

Response:

Based on this comment, we have added the details of between-group differences in FMD and serum ADMA levels, and results of periodontal status in the Abstract.

For example,

“(mean difference, −0.2%; 95% CI, −1.4–0.9; p = 0.708)” (Line 46).

“(mean difference, 0.01 nmol/L; 95% CI, −0.00–0.03; p = 0.122)” (Lines 46–47).

“Significant improvements in the average probing pocket depth were observed in the control and test groups. The bleeding on probing score in the test group was significantly reduced, while that in the control group was reduced, although not significantly” (Lines 47–50).

Comment:

-"The immune response following persistent bacteremia from periodontal lesions may lead to ACVD [3]. Furthermore, it has also been suggested that inflammatory mediators attributable to periodontal diseases are associated with ACVD [4-6]." Consequently, "bacteremia-induced ACVD routinely occurs even in the early stage of periodontal disease [2,19]". Undoubtedly, this would seem right. However, this would lead to the idea of a thorough eradication of all biofilms deemed responsible for infection, persistent bacteremia, and presence of inflammatory mediators (due to the biofilm). This refers to your "same-day full-mouth scaling, an ultrasonic scaler, along with hand instruments, and dental floss, and polishing paste with rotating rubber cup/brush", which is not deemed sufficient (even if widely established). Please refer to https://pubmed.ncbi.nlm.nih.gov/34269042/; there you will see that "(not only supragingival) biofilm removal by means of air polishing combined with low-abrasive erythritol has been shown to be more efficacious than the traditional polishing method". Thus, your applied methodology should be discussed more thoroughly, in particular since "thorough biofilm removal clearly results in delayed plaque regrowth rates", thus potentially showing a clear influence on inflammatory mediators and ACVD, in particular in combination with your "extra intervention in addition to standard self-care". Again, go to https://doi.org/10.3290/j.qi.b1763661, and discuss, both with regard to general effects and with regard to possible limitations.

Response:

Thank you for bringing this recent paper to our attention. The paper shows that “polishing combined with low-abrasive erythritol” is more efficacious than “air polishing or rubber cups with prophylaxis paste” for biofilm removal and delayed biofilm regrowth rates within 24 hours. As mentioned by Reviewer #2, the outcomes may be affected by the method employed for biofilm removal.

Based on this comment, we have added the following sentence to the revised manuscript with an appropriate citation:

“The main contents of standard care at the clinic are biofilm removal. Employment of more effective methods for biofilm removal may affect the improving FMD through delayed biofilm regrowth rates [28].” (Lines 325–327).

Comment:

- "We considered a 70% implementation rate as successful advanced periodontal self-care." Do you mean "successfully advanced periodontal self-care"?

Response:

Yes, we do. We decided that a “70% implementation rate represented successful advanced periodontal self-care” before the trial began. The decision was made in consideration of the burden on the subject and our empirical tolerance level for "successfully advanced periodontal self-care.” Those subjects with a below “70% implementation rate” were excluded from per-protocol analysis.

Comment:

- "(1–29 for smokers and 1–81 for nonsmokers)" should read "(1 to 29 for smokers and 1 to 81 for nonsmokers)".

Response:

Based on this comment, we have replaced “1–29 for smokers and 1–81 for nonsmokers” with “1 to 29 for smokers and 1 to 81 for nonsmokers” in the revised manuscript (Line 182).

Comment:

- "All the researchers confirmed that no one could identify the number." This would not seem clear, please clarify.

Response:

Based on this comment, we have added “from the outside of each sealed envelope” (Line 183).

Comment:

- "(Unexef18G, UNEX, Nagoya, Japan)" must read "(Unexef18G; UNEX, Nagoya, Japan)". Double check thoroughly.

Response:

Based on this comment, we have replaced “(Unexef18G, UNEX, Nagoya, Japan)” with “Unexef18G; UNEX, Nagoya, Japan” in the revised manuscript (Lines 203–204).

Comment:

- With your results section, please add exact p values. (See, for example, "Significant improvements in the average PPD were observed in the control and test groups. The BOP score in the test group was significantly reduced, while that in the control group was reduced, although not significantly.").

Response:

We have added a revised Table 3 and S2 Table based on the other reviewer’s comment. We have shown the exact p values in the Tables. We have removed statistical results from the text to avoid duplicate descriptions in the revised manuscript.

Comment:

- Again, with the first paragraph of your Disc section, please refer explicitly to H0, to provide a clear answer. "The findings were similar in the intention-to-treat and the per-protocol analysis." should read ""The findings were similar in the intention-to-treat and the per-protocol analysis; thus H0 was not rejected.

Response:

Thank you for your observation. Based on this comment, we have added the following sentence:

“Therefore, our null hypothesis ‘there was no difference in FMD between advanced self-care and standard care in patients with early-stage periodontal disease’ was not rejected.” (Lines 304–305).

Comment:

- "ADMA is a naturally occurring endogenous inhibitor of NO synthase." Reference(s) missing.

Response:

Based on this comment, we have added the following three references:

29. Najbauer J, Johnson BA, Young AL, Aswad DW. Peptides with sequences similar to glycine, arginine-rich motifs in proteins interacting with RNA are efficiently recognized by methyltransferase(s) modifying arginine in numerous proteins. J Biol Chem. 1993; 268(14): 10501-10509. PMID: 7683681.

30. Tang J, Kao PN, Herschman HR. Protein-arginine methyltransferase I, the predominant protein-arginine methyltransferase in cells, interacts with and is regulated by interleukin enhancer-binding factor 3. J Biol Chem. 2000; 275(26): 19866-19876. https://doi.org/10.1074/jbc.M000023200 PMID: 10749851.

31. MacAllister RJ, Fickling SA, Whitley GS, Vallance P. Metabolism of methylarginines by human vasculature; implications for the regulation of nitric oxide synthesis. Br J Pharmacol. 1994; 112(1): 43-48. https://doi.org/10.1111/j.1476-5381.1994.tb13026.x PMID: 7518309.

Comment:

- Same with "It reduces NO production; consequently, it can lead to endothelial dysfunction and cardiovascular events." Please revise carefully, and double check your whole draft, to support statements.

Response:

Based on this comment, we have added the following four references:

24. Sibal L, Agarwal SC, Home PD, Boger RH. The role of asymmetric dimethylarginine (ADMA) in endothelial dysfunction and cardiovascular disease. Curr Cardiol Rev. 2010; 6(2): 82-90. https://doi.org/10.2174/157340310791162659 PMID: 21532773.

32. Achan V, Broadhead M, Malaki M, Whitley G, Leiper J, MacAllister R, et al. Asymmetric dimethylarginine causes hypertension and cardiac dysfunction in humans and is actively metabolized by dimethylarginine dimethylaminohydrolase. Arterioscler Thromb Vasc Biol. 2003; 23(8): 1455-1459. https://doi.org/10.1161/01.ATV.0000081742.92006.59 PMID: 12805079.

33. Kielstein JT, Impraim B, Simmel S, Bode-Böger SM, Tsikas D, Frölich JC, et al. Cardiovascular effects of systemic NO synthase inhibition with asymmetric dimethylarginine in humans. Circulation. 2004; 109(2): 172-177. https://doi.org/10.1161/01.CIR.0000105764.22626.B1 PMID: 14662708.

34. Fliser D. Asymmetric dimethylarginine (ADMA): the silent transition from an ‘uraemic toxin’ to a global cardiovascular risk molecule. Eur J Clin Invest. 2005; 35(2): 71-79. https://doi.org/10.1111/j.1365-2362.2005.01457.x PMID: 15667575.

Comment:

- "Further investigations may be required in healthy subjects." To study what kind of effects? Please clarify.

Response:

Based on this comment, we have replaced “Further investigations may be required in healthy subjects” with “The reason for the discrepancy between the trials may be the underlying factors characterizing the study population which may be affecting FMD and/or plasma ADMA levels. Further investigations may be required to identify the factors in healthy subjects.” (Lines 336–339).

Comment:

- "A longer period might be required to alter endothelial function in patients with mild-to-moderate periodontal diseases." Please provide reasons why you think that prolonged follow-up periods would change the outcome.

Response:

When a certain treatment does not give positive results in a short span of time, continuation of the treatment is one of the common options available to medical practitioners. A longer period of care may increase the effectiveness of treatment for persistent bacteremia from periodontal lesions in patients with mild-to-moderate periodontal diseases.

Based on this comment, we have added “, which enables further improvement in bacteremia,” (Line 364).

Comment:

- Regarding the reference list, again, revise for uniform formatting (see Guidelines). Format must be "Bürgers R, Eidt A, Frankenberger R, Rosentritt M, Schweikl H, Handel G, et al. The anti-adherence activity and bactericidal effect of microparticulate silver additives in composite resin materials. Arch Oral Biol. 2009; 54(6): 595-601. https://doi.org/10.1016/j.archoralbio.2009.03.004 PMID: 19375069". Please revise for spacebar use to separate year and volume. Provide issue numbers (if available). Revise for correct doi numbers, and provide PMID numbers.

Response:

Based on this comment, we have reviewed all the references and adjusted them according to the Guidelines.

Comment:

Again, this study would seem worth following after revision. Compliments to the authors!

Response:

Thank you for the encouragement. We hope that our explanations and revisions are satisfactory.

---

## [Decision Letter · Decision Letter 2]

27 Aug 2021

Effect of advanced periodontal self-care in patients with early-stage periodontal diseases on endothelial function: An open-label, randomized controlled trial

PONE-D-21-11565R2

Dear Dr. Murata,

After having double checked your revised and re-submitted paper, I am pleased to inform you that your manuscript has been judged scientifically suitable for publication and will be formally accepted for publication once it meets all outstanding technical requirements.

Kind regards, congratulations and compliments, and stay healthy

Andrej M Kielbassa, Prof. Dr. med. dent. Dr. h. c.

Academic Editor

PLOS ONE

Reviewers' comments:

Reviewer's Responses to Questions

**Comments to the Author**

1. If the authors have adequately addressed your comments raised in a previous round of review and you feel that this manuscript is now acceptable for publication, you may indicate that here to bypass the “Comments to the Author” section, enter your conflict of interest statement in the “Confidential to Editor” section, and submit your "Accept" recommendation.

Reviewer #1: All comments have been addressed

2. Is the manuscript technically sound, and do the data support the conclusions?

Reviewer #1: Yes

3. Has the statistical analysis been performed appropriately and rigorously? 

Reviewer #1: Yes

4. Have the authors made all data underlying the findings in their manuscript fully available?

Reviewer #1: Yes

5. Is the manuscript presented in an intelligible fashion and written in standard English?

Reviewer #1: Yes

6. Review Comments to the Author

Reviewer #1: After having updated, adapted, and polished their revised and re-submitted draft, this manuscript is ready to proceed.

7. PLOS authors have the option to publish the peer review history of their article (what does this mean?). If published, this will include your full peer review and any attached files.

Reviewer #1: No

---

## [Editor Report · Acceptance letter]

15 Sep 2021

PONE-D-21-11565R2 

Effect of advanced periodontal self-care in patients with early-stage periodontal diseases on endothelial function: An open-label, randomized controlled trial 

Dear Dr. Murata:

I'm pleased to inform you that your manuscript has been deemed suitable for publication in PLOS ONE. Congratulations! Your manuscript is now with our production department. 

Kind regards, 

on behalf of

Prof. Dr. med. dent. Dr. h. c. Andrej M Kielbassa 

Academic Editor

PLOS ONE